# Massively Multilingual Word Embeddings

## Abstract

We introduce new methods for estimating and evaluating embeddings of words in more than fifty languages in a single shared embedding space. Our estimation methods, **multiCluster** and **multiCCA**, use dictionaries and monolingual data; they do not require parallel data. Our new evaluation method, **multiQVEC-CCA**, is shown to correlate better than previous ones with two downstream tasks (text categorization and parsing). We also describe a web portal for evaluation that will facilitate further research in this area, along with open-source releases of all our methods.

## 1 Introduction

Vector-space representations of words are widely used in statistical models of natural language. In addition to improving the performance on standard monolingual NLP tasks, shared representation of words across languages offers intriguing possibilities (Klementiev et al., 2012). For example, in machine translation, translating a word never seen in parallel data may be overcome by seeking its vector-space neighbors, provided the embeddings are learned from both plentiful monolingual corpora and more limited parallel data. A second opportunity comes from transfer learning, in which models trained in one language can be deployed in other languages. While previous work has used hand-engineered features that are cross-linguistically stable as the basis model transfer (Zeman and Resnik, 2008; McDonald et al., 2011; Tsvetkov et al., 2014), automatically learned embeddings offer the promise of better generalization at lower cost (Klementiev et al., 2012; Hermann and Blunsom, 2014; Guo et al., 2016). We there-

fore conjecture that developing estimation methods for massively multilingual word embeddings (i.e., embeddings for words in a large number of languages) will play an important role in the future of multilingual NLP.

This paper builds on previous work in multilingual embeddings and makes the following contributions:

- We propose two dictionary-based methods—**multiCluster** and **multiCCA**—for estimating multilingual embeddings which only require monolingual data and pairwise parallel dictionaries, and use them to train embeddings in 59 languages for which these resources are available (§2). Parallel corpora are not required but can be used when available. We show that the proposed methods work well in some settings and evaluation metrics.

- We adapt QVEC (Tsvetkov et al., 2015)[1] to evaluating multilingual embeddings (multiQVEC). We also develop a new evaluation method **multiQVEC-CCA** which addresses a theoretical shortcoming of multiQVEC (§3). Compared to other intrinsic metrics used in the literature, we show that both multiQVEC and multiQVEC-CCA achieve better correlations with extrinsic tasks.

- We develop an easy-to-use **web portal**[2] for evaluating arbitrary multilingual embeddings using a suite of intrinsic and extrinsic metrics (§4). Together with the provided benchmarks, the evaluation portal will substantially facilitate future research in this area.

## 2 Estimating Multilingual Embeddings

Let $\mathcal{L}$ be a set of languages, and let $\mathcal{V}^m$ be the set of surface forms (word types) in $m \in \mathcal{L}$. Let

---

[1] A method for evaluating monolingual word embeddings.
[2] http://128.2.220.95/multilingual

$\mathcal{V} = \bigcup_{m \in \mathcal{L}} \mathcal{V}^m$. Our goal is to estimate a partial **embedding** function $E : \mathcal{L} \times \mathcal{V} \nrightarrow \mathbb{R}^d$ (allowing a surface form that appears in two languages to have different vectors in each). We would like to estimate this function such that: **(i)** semantically similar words in the same language are nearby, **(ii)** translationally equivalent words in different languages are nearby, and **(iii)** the domain of the function covers as many words in $\mathcal{V}$ as possible.

We use distributional similarity in a monolingual corpus $M^m$ to model semantic similarity between words in the same language. For cross-lingual similarity, either a parallel corpus $P^{m,n}$ or a bilingual dictionary $D^{m,n} \subset \mathcal{V}^m \times \mathcal{V}^n$ can be used. Our methods focus on the latter, in some cases extracting $D^{m,n}$ from a parallel corpus.[3]

Most previous work on multilingual embeddings only considered the bilingual case, $| \mathcal{L} |= 2$. We focus on estimating multilingual embeddings for $| \mathcal{L} |> 2$ and describe two novel dictionary-based methods (**multiCluster** and **multiCCA**). We then describe our baselines: a variant of Coulmance et al. (2015) and Guo et al. (2016) (henceforth referred to as multiSkip),[4] and the translation-invariance matrix factorization method (Gardner et al., 2015).

## 2.1 MultiCluster

In this approach, we decompose the problem into two simpler subproblems: $E = E_{\text{embed}} \circ E_{\text{cluster}}$, where $E_{\text{cluster}} : \mathcal{L} \times \mathcal{V} \nrightarrow \mathcal{C}$ deterministically maps words to multilingual clusters $\mathcal{C}$, and $E_{\text{embed}} : \mathcal{C} \rightarrow \mathbb{R}^d$ assigns a vector to each cluster. We use a bilingual dictionary to find clusters of translationally equivalent words, then use distributional similarities of the clusters in monolingual corpora from all languages in $\mathcal{L}$ to estimate an embedding for each cluster. By forcing words from different languages in a cluster to share the same embedding, we create anchor points in the vector space to bridge languages.

More specifically, we define the clusters as the

---

[3]To do this, we align the corpus using fast_align (Dyer et al., 2013) in both directions. The estimated parameters of the word translation distributions are used to select pairs: $D^{m,n} = \left\{ (u, v) \mid u \in \mathcal{V}^m, v \in \mathcal{V}^n, p_{m|n}(u \mid v) \times p_{n|m}(v \mid u) > \tau \right\}$, where the threshold $\tau$ trades off dictionary recall and precision. We fixed $\tau = 0.1$ early on based on manual inspection of the resulting dictionaries.

[4]We developed multiSkip independently of Coulmance et al. (2015) and Guo et al. (2016). One important distinction is that multiSkip is only trained on parallel corpora, while Coulmance et al. (2015) and Guo et al. (2016) also use monolingual corpora.

connected components in a graph where nodes are (language, surface form) pairs and edges correspond to translation entries in $D^{m,n}$. We assign arbitrary IDs to the clusters and replace each word token in each monolingual corpus with the corresponding cluster ID, and concatenate all modified corpora. The resulting corpus consists of multilingual cluster ID sequences. We can then apply any monolingual embedding estimator; here, we use the skipgram model from Mikolov et al. (2013a).

## 2.2 MultiCCA

Our proposed method (**multiCCA**) extends the bilingual embeddings of Faruqui and Dyer (2014). First, they use monolingual corpora to train monolingual embeddings for each language independently ($E^m$ and $E^n$), capturing semantic similarity within each language separately. Then, using a bilingual dictionary $D^{m,n}$, they use canonical correlation analysis (CCA) to estimate linear projections from the ranges of the monolingual embeddings $E^m$ and $E^n$, yielding a bilingual embedding $E^{m,n}$. The linear projections are defined by $T_{m \rightarrow m,n}$ and $T_{n \rightarrow m,n} \in \mathbb{R}^{d \times d}$; they are selected to maximize the correlation between $T_{m \rightarrow m,n} E^m(u)$ and $T_{n \rightarrow m,n} E^n(v)$ where $(u, v) \in D^{m,n}$. The bilingual embedding is then defined as $E_{\text{CCA}}(m, u) = T_{m \rightarrow m,n} E^m(u)$ (and likewise for $E_{\text{CCA}}(n, v)$).

In this work, we use a simple extension (in hindsight) to construct multilingual embeddings for more languages. We let the vector space of the initial (monolingual) English embeddings serve as the multilingual vector space (since English typically offers the largest corpora and wide availability of bilingual dictionaries). We then estimate projections from the monolingual embeddings of the other languages into the English space.

We start by estimating, for each $m \in \mathcal{L} \setminus \{\text{en}\}$, the two projection matrices: $T_{m \rightarrow m,\text{en}}$ and $T_{\text{en} \rightarrow m,\text{en}}$; these are guaranteed to be non-singular. We then define the multilingual embedding as $E_{\text{CCA}}(\text{en}, u) = E^{\text{en}}(u)$ for $u \in \mathcal{V}^{\text{en}}$, and $E_{\text{CCA}}(m, v) = T_{\text{en} \rightarrow m,\text{en}}^{-1} T_{m \rightarrow m,\text{en}} E^m(v)$ for $v \in \mathcal{V}^m, m \in \mathcal{L} \setminus \{\text{en}\}$.

## 2.3 MultiSkip

Luong et al. (2015b) proposed a method for estimating bilingual embeddings which only makes use of parallel data; it extends the skipgram model of Mikolov et al. (2013a). The skipgram model defines a distribution over words $u$ that occur in a

context window (of size $K$) of a word $v$:

$$p(u \mid v) = \frac{\exp E_{\text{skipgram}}(m,v)^\top E_{\text{context}}(m,u)}{\sum_{u' \in \mathcal{V}^m} \exp E_{\text{skipgram}}(m,v)^\top E_{\text{context}}(m,u')}$$

In practice, this distribution can be estimated using a noise contrastive estimation approximation (Gutmann and Hyvärinen, 2012) while maximizing the log-likelihood:

$$\sum_{i \in \text{pos}(M^m)} \sum_{k \in \{-K,\dots,-1,1,\dots,K\}} \log p(u_{i+k} \mid u_i)$$

where $\text{pos}(M^m)$ are the indices of words in the monolingual corpus $M^m$.

To establish a bilingual embedding, with a parallel corpus $P^{m,n}$ of source language $m$ and target language $n$, Luong et al. (2015b) estimate conditional models of words in both source and target positions. The source positions are selected as sentential contexts (similar to monolingual skipgram), and the bilingual contexts come from aligned words. The bilingual objective is to maximize:

$$\sum_{i \in m\text{-pos}(P_{m,n})} \sum_{k \in \{-K,\dots,-1,1,\dots,K\}} \frac{\log p(u_{i+k} \mid u_i)}{+ \log p(v_{a(i)+k} \mid u_i)}$$

$$+ \sum_{j \in n\text{-pos}(P_{m,n})} \sum_{k \in \{-K,\dots,-1,1,\dots,K\}} \frac{\log p(v_{j+k} \mid v_j)}{+ \log p(u_{a(j)+k} \mid v_j)}$$

where $m\text{-pos}(P_{m,n})$ and $n\text{-pos}(P_{m,n})$ are the indeces of the source and target tokens in the parallel corpus respectively, $a(i)$ and $a(j)$ are the positions of words that align to $i$ and $j$ in the other language. It is easy to see how this method can be extended for more than two languages by summing up the bilingual objectives for all available parallel corpora.

### 2.4 Translation-invariance

Gardner et al. (2015) proposed that multilingual embeddings should be translation invariant. Consider a matrix $X \in \mathbb{R}^{|\mathcal{V}| \times |\mathcal{V}|}$ which summarizes the pointwise mutual information statistics between pairs of words in monolingual corpora, and let $UV^\top$ be a low-rank decomposition of $X$ where $U, V \in \mathbb{R}^{|\mathcal{V}| \times d}$. Now, consider another matrix $A \in \mathbb{R}^{|\mathcal{V}| \times |\mathcal{V}|}$ which summarizes bilingual alignment frequencies in a parallel corpus. Gardner et al. (2015) solves for a low-rank decomposition $UV^\top$ which both approximates $X$ as well as its

transformations $A^\top X$, $XA$ and $A^\top XA$ by defining the following objective:

$$min_{U,V} \|X - UV^\top\|^2 + \|XA - UV^\top\|^2$$
$$+ \|A^\top X - UV^\top\|^2 + \|A^\top XA - UV^\top\|^2$$

The multilingual embeddings are then taken to be the rows of the matrix $U$.

## 3 Evaluating Multilingual Embeddings

One of our contributions is to streamline the evaluation of multilingual embeddings. In addition to assessing goals (i–iii) stated in §2, a good evaluation metric should also **(iv)** show good correlation with performance in downstream applications and **(v)** be computationally efficient.

It is easy to evaluate the coverage (iii) by counting the number of words covered by an embedding function in a closed vocabulary. Intrinsic evaluation metrics are generally designed to be computationally efficient (v) but may or may not meet the goals (i, ii, iv). Although intrinsic evaluations will never be perfect, a standard set of evaluation metrics will help drive research. By design, standard (monolingual) word similarity tasks meet (i) while cross-lingual word similarity tasks and the word translation tasks meet (ii). We propose another evaluation method (multiQVEC-CCA), designed to simultaneously assess goals (i, ii). MultiQVEC-CCA extends QVEC (Tsvetkov et al., 2015), a recently proposed monolingual evaluation method, addressing fundamental flaws and extending it to multiple languages. To assess the degree to which these evaluation metrics meet (iv), in §5 we perform a correlation analysis looking at which intrinsic metrics are best correlated with downstream task performance—i.e., we evaluate the evaluation metrics.

### 3.1 Word similarity

Word similarity datasets such as WordSim-353 (Agirre et al., 2009) and MEN (Bruni et al., 2014) provide human judgments of semantic similarity. By ranking words by cosine similarity and by their empirical similarity judgments, a ranking correlation can be computed that assesses how well the estimated vectors capture human intuitions about semantic relatedness.

Some previous work on bilingual and multilingual embeddings focuses on monolingual word similarity to evaluate embeddings (e.g., Faruqui and Dyer, 2014). This approach is limited because

it cannot measure the degree to which embeddings from different languages are similar (ii). For this paper, we report results on an English word similarity task, the Stanford RW dataset (Luong et al., 2013), as well as a combination of several cross-lingual word similarity datasets (Camacho-Collados et al., 2015).

### 3.2 Word translation

This task directly assesses the degree to which translationally equivalent words in different languages are nearby in the embedding space. The evaluation data consists of word pairs which are known to be translationally equivalent. The score for one word pair $(l_1, w_1), (l_2, w_2)$ both of which are covered by an embedding $E$ is 1 if $\text{cosine}(E(l_1, w_1), E(l_2, w_2)) \geq \text{cosine}(E(l_1, w_1), E(l_2, w_2')) \, \forall \, w_2' \in G^{l_2}$ where $G^{l_2}$ is the set of words of language $l_2$ in the evaluation dataset, and cosine is the cosine similarity function. Otherwise, the score for this word pair is 0. The overall score is the average score for all word pairs covered by the embedding function. This is a variant of the method used by Mikolov et al. (2013b) to evaluate bilingual embeddings.

### 3.3 Correlation-based evaluation

We introduce QVEC-CCA—an intrinsic evaluation measure of the quality of word embeddings. Our method is an improvement of QVEC—a monolingual evaluation based on alignment of embeddings to a matrix of features extracted from a linguistic resource (Tsvetkov et al., 2015). We review QVEC, and then describe QVEC-CCA.

**QVEC.** The main idea behind QVEC is to quantify the linguistic content of word embeddings by maximizing the correlation with a manually-annotated linguistic resource. Let the number of common words in the vocabulary of the word embeddings and the linguistic resource be $N$. To quantify the semantic content of embeddings, a semantic linguistic matrix $\mathbf{S} \in \mathbb{R}^{P \times N}$ is constructed from a semantic database, with a column vector for each word. Each word vector is a distribution of the word over $P$ linguistic properties, based on annotations of the word in the database. Let $\mathbf{X} \in \mathbb{R}^{D \times N}$ be embedding matrix with every row as a dimension vector $\mathbf{x} \in \mathbb{R}^{1 \times N}$. $D$ denotes the dimensionality of word embeddings. Then, $\mathbf{S}$ and $\mathbf{X}$ are aligned to maximize the cumulative correlation between the aligned dimensions of the two

matrices. Specifically, let $\mathbf{A} \in \{0, 1\}^{D \times P}$ be a matrix of alignments such that $a_{ij} = 1$ iff $\mathbf{x}_i$ is aligned to $\mathbf{s}_j$, otherwise $a_{ij} = 0$. If $r(\mathbf{x}_i, \mathbf{s}_j)$ is the Pearson's correlation between vectors $\mathbf{x}_i$ and $\mathbf{s}_j$, then QVEC is defined as:

$$\text{QVEC} = max_{\mathbf{A}: \sum_j a_{ij} \leq 1} \sum_{i=1}^{X} \sum_{j=1}^{S} r(\mathbf{x}_i, \mathbf{s}_j) \times a_{ij}$$

The constraint $\sum_j a_{ij} \leq 1$, warrants that one distributional dimension is aligned to at most one linguistic dimension.

QVEC has been shown to correlate strongly with downstream semantic tasks (Tsvetkov et al., 2015). However, it suffers from two major weaknesses. First, it is not invariant to linear transformations of the embeddings' basis, whereas the bases in word embeddings are generally arbitrary (Szegedy et al., 2014). Second, a sum of correlations produces an unnormalized score: the more dimensions in the embedding matrix the higher the score. This precludes comparison of models of different dimensionality. QVEC-CCA simultaneously addresses both problems.

**QVEC-CCA.** To measure correlation between the embedding matrix $\mathbf{X}$ and the linguistic matrix $\mathbf{S}$, instead of cumulative dimension-wise correlation we employ CCA. CCA finds two sets of basis vectors, one for $\mathbf{X}^\top$ and the other for $\mathbf{S}^\top$, such that the correlations between the projections of the matrices onto these basis vectors are maximized. Formally, CCA finds a pair of basis vectors $\mathbf{v}$ and $\mathbf{w}$ such that

$$\text{QVEC-CCA} = \text{CCA}(\mathbf{X}^\top, \mathbf{S}^\top)$$
$$= max_{\mathbf{v}, \mathbf{w}} \, r(\mathbf{X}^\top \mathbf{v}, \mathbf{S}^\top \mathbf{w})$$

Thus, QVEC-CCA ensures invariance to the matrices bases rotation, and since it is a single correlation, it produces a score in $[-1, 1]$. Both QVEC and QVEC-CCA rely on a matrix of linguistic properties constructed from a manually crafted linguistic resource. We extend both methods to multilingual evaluations—multiQVEC and multiQVEC-CCA—by constructing the linguistic matrix using supersense tag annotations for English (Miller et al., 1993), Danish (Martínez Alonso et al., 2015; Martínez Alonso et al., 2016) and Italian (Montemagni et al., 2003).

### 3.4 Extrinsic tasks

In order to evaluate how useful the word embeddings are for a downstream task, we use the em-

bedding vector as a dense feature representation of each word in the input, and deliberately remove any other feature available for this word (e.g., prefixes, suffixes, part-of-speech). For each task, we train one model on the aggregate training data available for several languages, and evaluate on the aggregate evaluation data in the same set of languages. We apply this for multilingual document classification and multilingual dependency parsing.

For document classification, we follow Klementiev et al. (2012) in using the RCV corpus of newswire text, and train a classifier which differentiates between four topics. While most previous work used this data only in a bilingual setup, we simultaneously train the classifier on documents in seven languages,[5] and evaluate on the development/test section of those languages. For this task, we report the average classification accuracy on the test set.

For dependency parsing, we train the stack-LSTM parser of Dyer et al. (2015) on a subset of the languages in the universal dependencies v1.1,[6] and test on the same languages, reporting unlabeled attachment scores. We remove all part-of-speech and morphology features from the data, and prevent the model from optimizing the word embeddings used to represent each word in the corpus, thereby forcing the parser to rely completely on the provided (pretrained) embeddings as the token representation. Although omitting other features (e.g., parts of speech) hurts the performance of the parser, it emphasizes the contribution of the word embeddings being studied.

## 4 Evaluation Portal

In order to facilitate future research on multilingual word embeddings, we developed a web portal to enable researchers who develop new estimation methods to evaluate them using a suite of evaluation tasks. The portal serves the following purposes:

- Download the monolingual and bilingual data we used to estimate multilingual embeddings in this paper,

- Download standard development/test data sets for each of the evaluation metrics to help re-

searchers working in this area report trustworthy and replicable results,[7]

- Upload arbitrary multilingual embeddings, scan which languages are covered by the embeddings, allow the user to pick among the compatible evaluation tasks, and receive evaluation scores for the selected tasks, and

- Register a new evaluation data set or a new evaluation metric via the github repository which mirrors the backend of the web portal.

## 5 Experiments

Our experiments are designed to show two primary sets of results: (i) how well the proposed intrinsic evaluation metrics correlate with downstream tasks (§5.1) and (ii) which estimation methods work best according to each metric (§5.2). The data used for training and evaluation are available for download on the evaluation portal.

### 5.1 Correlations between intrinsic vs. extrinsic evaluation metrics

In this experiment, we consider four intrinsic evaluation metrics (cross-lingual word similarity, word translation, multiQVEC and multiQVEC-CCA) and two extrinsic evaluation metrics (multilingual document classification and multilingual parsing).

**Data:** For the cross-lingual word similarity task, we use disjoint subsets of the en-it MWS353 dataset (Leviant and Reichart, 2015) for development (308 word pairs) and testing (307 word pairs). For the word translation task, we use Wiktionary to extract a development set (647 translations) and a test set (647 translations) of translationally-equivalent word pairs in en-it, en-da and da-it. For both multiQVEC and multiQVEC-CCA, we used disjoint subsets of the multilingual (en, da, it) supersense tag annotations described in §3 for development (12,513 types) and testing (12,512 types).

For the document classification task, we use the multilingual RCV corpus (en, it, da). For the dependency parsing task, we use the universal dependencies v1.1 (Agić et al., 2015) in three languages (en, da, it).

---

| ($\rightarrow$) extrinsic task ($\downarrow$) intrinsic metric | document classification | dependency parsing |
|---|---|---|
| word similarity | 0.386 | 0.007 |
| word translation | 0.066 | -0.292 |
| multiQVEC | 0.635 | **0.444** |
| multiQVEC-CCA | **0.896** | 0.273 |

Table 1: Correlations between intrinsic evaluation metrics (rows) and downstream task performance (columns).

**Setup:** To estimate correlations between the proposed intrinsic evaluation metrics and downstream task performance, we train a total of 17 different multilingual embeddings for three languages (English, Italian and Danish). To compute the correlations, we evaluate each of the 17 embeddings (12 multiCluster embeddings, 1 multiCCA embeddings, 1 multiSkip embeddings, 2 translation-invariance embeddings) according to each of the six evaluation metrics (4 intrinsic, 2 extrinsic).[8]

**Results:** Table 1 shows Pearson's correlation coefficients of eight (intrinsic metric, extrinsic metric) pairs. Although each of two proposed methods multiQVEC and multiQVEC-CCA correlate better with a different extrinsic task, we establish (i) that intrinsic methods previously used in the literature (cross-lingual word similarity and word translation) correlate poorly with downstream tasks, and (ii) that the intrinsic methods proposed in this paper (multiQVEC and multiQVEC-CCA) correlate better with both downstream tasks, compared to cross-lingual word similarity and word translation.[9]

## 5.2 Evaluating multilingual estimation methods

We now turn to evaluating the four estimation methods described in §2. We use the proposed methods (i.e., multiCluster and multiCCA) to

---

[8]The 102 ($17 \times 6$) values used to compute Pearson's correlation coefficient are provided in the supplementary material.

[9]Although supersense annotations exist for other languages, the annotations are inconsistent across languages and may not be publicly available, which is a disadvantage of the multiQVEC and multiQVEC-CCA metrics. Therefore, we recommend that future multilingual supersense annotation efforts use the same set of supersense tags used in other languages. If the word embeddings are primarily needed for encoding syntactic information, one could use tag dictionaries based on the universal POS tag set (Petrov et al., 2012) instead of supersense tags.

| Task | multiCluster | multiCCA |
|---|---|---|
| dependency parsing | 48.4 [72.1] | **48.8** [69.3] |
| doc. classification | 90.3 [52.3] | **91.6** [52.6] |
| mono. wordsim | 14.9 [71.0] | **43.0** [71.0] |
| cross. wordsim | 12.8 [78.2] | **66.8** [78.2] |
| word translation | 30.0 [38.9] | **83.6** [31.8] |
| mono. QVEC | 7.6 [99.6] | **10.7** [99.0] |
| multiQVEC | 8.3 [86.4] | **8.7** [87.0] |
| mono. QVEC-CCA | 53.8 [99.6] | **63.4** [99.0] |
| multiQVEC-CCA | 37.4 [86.4] | **42.0** [87.0] |

Table 2: Results for multilingual embeddings that cover 59 languages. Each row corresponds to one of the embedding evaluation metrics we use (higher is better). Each column corresponds to one of the embedding estimation methods we consider; i.e., numbers in the same row are comparable. Numbers in square brackets are coverage percentages.

train multilingual embeddings in 59 languages for which bilingual translation dictionaries are available.[10] In order to compare our methods to baselines which use parallel data (i.e., multiSkip and translation-invariance), we also train multilingual embeddings in a smaller set of 12 languages for which high-quality parallel data are available.[11]

**Training data:** We use Europarl en-xx parallel data for the set of 12 languages. We obtain en-xx bilingual dictionaries from two different sources. For the set of 12 languages, we extract the bilingual dictionaries from the Europarl parallel corpora. For the remaining 47 languages, dictionaries were formed by translating the 20k most common words in the English monolingual corpus with Google Translate, ignoring translation pairs with identical surface forms and multi-word translations.

**Evaluation data:** Monolingual word similarity uses the MEN dataset in Bruni et al. (2014) as a development set and Stanford's Rare Words dataset in Luong et al. (2013) as a test set. For the cross-lingual word similarity task, we aggregate the RG-65 datasets in six language pairs (fr-es, fr-de, en-fr, en-es, en-de, de-es). For the word translation

---

[10]The 59-language set is { bg, cs, da, de, el, en, es, fi, fr, hu, it, sv, zh, af, ca, iw, cy, ar, ga, zu, et, gl, id, ru, nl, pt, la, tr, ne, lv, lt, tg, ro, is, pl, yi, be, hy, hr, jw, ka, ht, fa, mi, bs, ja, mg, tl, ms, uz, kk, sr, mn, ko, mk, so, uk, sl, sw }.

[11]The 12-language set is {bg, cs, da, de, el, en, es, fi, fr, hu, it, sv}.

task, we use Wiktionary to extract translationally-equivalent word pairs to evaluate multilingual embeddings for the set of 12 languages. Since Wiktionary-based translations do not cover all 59 languages, we use Google Translate to obtain en-xx bilingual dictionaries to evaluate the embeddings of 59 languages. For QVEC and QVEC-CCA, we split the English supersense annotations used in Tsvetkov et al. (2015) into a development set and a test set. For multiQVEC and multiQVEC-CCA, we use supersense annotations in English, Italian and Danish. For the document classification task, we use the multilingual RCV corpus in seven languages (da, de, en, es, fr, it, sv). For the dependency parsing task, we use the universal dependencies v1.1 in twelve languages (bg, cs, da, de, el, en, es, fi, fr, hu, it, sv).

**Setup:** All word embeddings in the following results are 512-dimensional vectors. Methods which indirectly use skipgram (i.e., multiCCA, multiSkip, and multiCluster) are trained using 10 epochs of stochastic gradient descent, and use a context window of size 5. The translation-invariance method use a context window of size 3.[12] We only estimate embeddings for words/clusters which occur 5 times or more in the monolingual corpora. In a postprocessing step, all vectors are normalized to unit length. Multi-Cluster uses a maximum cluster size of 1,000 and 10,000 for the set of 12 and 59 languages, respectively. In the English tasks (monolingual word similarity, QVEC, QVEC-CCA), skipgram embeddings (Mikolov et al., 2013a) and multiCCA embeddings give identical results (since we project words in other languages to the English vector space, estimated using the skipgram model). The software used to train all embeddings as well as the trained embeddings are available for download on the evaluation portal.[13]

We note that intrinsic evaluation of word embeddings (e.g., word similarity) typically ignores test instances which are not covered by the embeddings being studied. When the vocabulary used in two sets of word embeddings is different, which is often the case, the intrinsic evaluation score for each set may be computed based on a different set

of test instances, which may bias the results in unexpected ways. For instance, if one set of embeddings only covers frequent words while the other set also covers infrequent words, the scores of the first set may be inflated because frequent words appear in many different contexts and are therefore easier to estimate than infrequent words. To partially address this problem, we report the coverage of each set of embeddings in square brackets. When the difference in coverage is large, we repeat the evaluation using only the intersection of vocabularies covered by all embeddings being evaluated. Extrinsic evaluations are immune to this problem because the score is computed based on all test instances regardless of the coverage.

**Results [59 languages].** We train the proposed dictionary-based estimation methods (multiCluster and multiCCA) for 59 languages, and evaluate the trained embeddings according to nine different metrics in Table 2. The results show that, when trained on a large number of languages, multiCCA consistently outperforms multiCluster according to all evaluation metrics. Note that most differences in coverage between multiCluster and multiCCA are relatively small.

It is worth noting that the mainstream approach of estimating one vector representation per word type (rather than word token) ignores the fact that the same word may have different semantics in different contexts. The multiCluster method exacerbates this problem by estimating one vector representation per cluster of translationally equivalent words. The added semantic ambiguity severely hurts the performance of multiCluster with 59 languages, but it is still competitive with 12 languages (see below).

**Results on [12 languages].** We compare the proposed dictionary-based estimation methods to parallel text-based methods in Table 3. The ranking of the four estimation methods is not consistent across all evaluation metrics. This is unsurprising since each metric evaluates different traits of word embeddings, as detailed in §3. However, some patterns are worth noting in Table 3.

In five of the evaluations (including both extrinsic tasks), the best performing method is a dictionary-based one proposed in this paper. In the remaining four intrinsic methods, the best performing method is the translation-invariance method. MultiSkip ranks last in five evaluations,

---

[12]Training translation-invariance embeddings with larger context window sizes using the matlab implementation provided by Gardner et al. (2015) is computationally challenging.

[13]URLs to software libraries on Github are redacted to comply with the double-blind reviewing of CoNLL.

| | Task | multiCluster | multiCCA | multiSkip | invariance |
|---|---|---|---|---|---|
| extrinsic metrics | dependency parsing | **61.0** [70.9] | 58.7 [69.3] | 57.7 [68.9] | 59.8 [68.6] |
| | document classification | **92.1** [48.1] | **92.1** [62.8] | 90.4 [45.7] | 91.1 [31.3] |
| intrinsic metrics | monolingual word similarity | 38.0 [57.5] | 43.0 [71.0] | 33.9 [55.4] | **51.0** [23.0] |
| | multilingual word similarity | 58.1 [74.1] | **66.6** [78.2] | 59.5 [67.5] | 58.7 [63.0] |
| | word translation | 43.7 [45.2] | 35.7 [53.2] | 46.7 [39.5] | **63.9** [30.3] |
| | monolingual QVEC | 10.3 [98.6] | **10.7** [99.0] | 8.4 [98.0] | 8.1 [91.7] |
| | multiQVEC | **9.3** [82.0] | 8.7 [87.0] | 8.7 [87.0] | 5.3 [74.7] |
| | monolingual QVEC-CCA | 62.4 [98.6] | 63.4 [99.0] | 58.9 [98.0] | **65.8** [91.7] |
| | multiQVEC-CCA | 43.3 [82.0] | 41.5 [87.0] | 36.3 [75.6] | **46.2** [74.7] |

Table 3: Results for multilingual embeddings that cover Bulgarian, Czech, Danish, Greek, English, Spanish, German, Finnish, French, Hungarian, Italian and Swedish. Each row corresponds to one of the embedding evaluation metrics we use (higher is better). Each column corresponds to one of the embedding estimation methods we consider; i.e., numbers in the same row are comparable. Numbers in square brackets are coverage percentages.

and never ranks first. Since our implementation of multiSkip does not make use of monolingual data, it only learns from monolingual contexts observed in parallel corpora, it misses the opportunity to learn from contexts in the much larger monolingual corpora. Trained for 12 languages, multiCluster is competitive in four evaluations (and ranks first in three).

We note that multiCCA consistently achieves better coverage than the translation-invariance method. For intrinsic measures, this confounds the performance comparison. A partial solution is to test only on word types for which all four methods have a vector; this subset is in no sense a representative sample of the vocabulary. In this comparison (provided in the supplementary material), we find a similar pattern of results, though multiCCA outperforms the translation-invariance method on the monolingual word similarity task. Also, the gap (between multiCCA and the translation-invariance method) reduces to 0.7 in monolingual QVEC-CCA and 2.5 in multiQVEC-CCA.

## 6 Related Work

There is a rich body of literature on bilingual embeddings, including work on machine translation (Zou et al., 2013; Hermann and Blunsom, 2014; Cho et al., 2014; Luong et al., 2015b; Luong et al., 2015a, *inter alia*),[14] cross-lingual dependency parsing (Guo et al., 2015; Guo et al., 2016), and cross-lingual document classification (Klementiev

et al., 2012; Gouws et al., 2014; Kociský et al., 2014). Al-Rfou' et al. (2013) trained word embeddings for more than 100 languages, but the embeddings of each language are trained independently (i.e., embeddings of words in different languages do not share the same vector space). Word clusters are a related form of distributional representation; in clustering, cross-lingual distributional representations were proposed as well (Och, 1999; Täckström et al., 2012). Haghighi et al. (2008) used CCA to learn bilingual lexicons from monolingual corpora.

## 7 Conclusion

We proposed two dictionary-based estimation methods for multilingual word embeddings, **multiCCA** and **multiCluster**, and used them to train embeddings for 59 languages. We characterized important shortcomings of the QVEC previously used to evaluate monolingual embeddings, and proposed an improved metric **multiQVEC-CCA**. Both multiQVEC and multiQVEC-CCA obtain better correlations with downstream tasks compared to intrinsic methods previously used in the literature. Finally, in order to help future research in this area, we created a **web portal** for users to upload their multilingual embeddings and easily evaluate them on nine evaluation metrics, with two modes of operation (development and test) to encourage sound experimentation practices.

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
