# Peer review of "Massively Multilingual Word Embeddings"

_CoNLL 2016 — decision unknown_

[Official Review · Reviewer 1 · rating 3 · confidence 4]
soundness 4 · originality 3 · clarity 4 · impact 3 · substance 3 · appropriateness 4 · meaningful comparison 3 · replicability 5 · presentation format Poster

This paper describes four methods of obtaining multilingual word embeddings and
a modified QVEC metric for evaluating the efficacy of these embeddings. The
embedding methods are: 

(1) multiCluster : Uses a dictionary to map words to multilingual clusters.
Cluster embeddings are then obtained which serve as embeddings for the words
that reside in each cluster. 

(2) multiCCA : Extends the approach presented by Faruqui and Dyer (2014) for
embedding bilingual words, to multilingual words by using English embeddings as
the anchor space. Bilingual dictionaries (other_language -> English) are then
used to obtain projections from other monolingual embeddings for words in other
languages to the anchor space. 

(3) multiSkip : Extends the approach presented by Luong et al. (2015b) for
embedding using source and target context (via alignment), to the multilingual
case by extending the objective function to include components for all
available parallel corpora. 

(4) Translation invariance : Uses a low rank decomposition of the word PMI
matrix with an objective with includes bilingual alignment frequency
components. May only work for  bilingual embeddings. 

The evaluation method uses CCA to maximize the correlation between the word
embeddings and possibly hand crafted linguistic data. Basis vectors are
obtained for the aligned dimensions which produce a score which is invariant to
rotation and linear transformations. The proposed method also extends this to
multilingual evaluations. 

In general, the paper is well written and describes the work clearly. A few
major issues:

(1) What is the new contribution with respect to the translation invariance
embedding approach of Gardner et al.? If it is the extension to multilingual
embeddings, a few lines explaining the novelty would help. 

(2) The use of super-sense annotations across multiple languages is a problem.
The number of features in the intersection of multiple languages may become
really small. How do the authors propose to address this problem (beyond
footnote 9)?

(3) How much does coverage affect the score in table 2? For example, for
dependency parsing, multi cluster and multiCCA have significantly different
coverage numbers with scores that are close. 

(4) In general, the results in table 3 do not tell a consistent story. Mainly,
for most of the intrinsic metrics, the multilingual embedding techniques do not
seem to perform the best.  Given that one of the primary goals of this paper
was to create embeddings that perform well under the word translation metric
(intra-language), it is disappointing that the method that performs best (by
far) is the invariance approach. It is also strange that the multi-cluster
approach, which discards inter-cluster (word and language) semantic information
performs the best with respect to the extrinsic metrics.

Other questions for the authors:

(1) What is the loss in performance by fixing the word embeddings in the
dependency parsing task? What was the gain by simply using these embeddings as
alternatives to the random embeddings in the LSTM stack parser? 

(2) Is table 1 an average over the 17 embeddings described in section 5.1? 

(3) Are there any advantages of using the multi-Skip approach instead of
learning bilingual embeddings and performing multi-CCA to learning projections
across the distinct spaces?

(4) The dictionary extraction approach (from parallel corpora via alignments or
from google translate) may not reflect the challenges of using real lexicons.
Did you explore the use of any real multi-lingual dictionaries?

[Official Review · Reviewer 2 · rating 3 · confidence 4]
soundness 5 · originality 3 · clarity 5 · impact 4 · substance 5 · appropriateness 4 · meaningful comparison 5 · replicability 5 · presentation format Poster

This paper proposes two dictionary-based methods for estimating multilingual
word embeddings, one motivated in clustering (MultiCluster) and another in
canonical correlation analysis (MultiCCA).
In addition, a supersense similarity measure is proposed that improves on QVEC
by substituting its correlation component with CCA, and by taking into account
multilingual evaluation.
 The evaluation is performed on a wide range of tasks using the web portal
developed by the authors; it is shown that in some cases the proposed
representation methods outperform two other baselines.

I think the paper is very well written, and represents a substantial amount of
work done. The presented representation-learning and evaluation methods are
certainly timely. I also applaud the authors for the meticulous documentation.

My general feel about this paper, however, is that it goes (perhaps) in too
much breadth at the expense of some depth. I'd prefer to see a thorougher
discussion of results (e.g. regarding the conflicting outcome for MultiCluster
between 59- and 12-language set-up; regarding the effect of estimation
parameters and decisions in MultiCluster/CCA). So, while I think the paper is
of high practical value to me and the research community (improved QVEC
measure, web portal), I frankly haven't learned that much from reading it, i.e.
in terms of research questions addressed and answered.

Below are some more concrete remarks.

It would make sense to include the correlation results (Table 1) for
monolingual QVEC and QVEC-CCA as well. After all, it is stated in l.326--328
that the proposed QVEC-CCA is an improvement over QVEC.

Minor:
l. 304: "a combination of several cross-lingual word similarity datasets" ->
this sounds as though they are of different nature, whereas they are really of
the same kind, just different languages, right?

p. 3: two equations exceed the column margin

Lines 121 and 147 only mention Coulmance et al and Guo et al when referring to
the MultiSkip baseline, but section 2.3 then only mentions Luong et al. So,
what's the correspondence between these works?

While I think the paper does reasonable justice in citing the related works,
there are more that are relevant and could be included:

Multilingual embeddings and clustering:
Chandar A P, S., Lauly, S., Larochelle, H., Khapra, M. M., Ravindran, B.,
Raykar, V. C., and Saha, A. (2014). An autoencoder approach to learning
bilingual word representations. In NIPS.
Hill, F., Cho, K., Jean, S., Devin, C., and Bengio, Y. (2014). Embedding word
similarity with neural machine translation. arXiv preprint arXiv:1412.6448.
Lu, A., Wang, W., Bansal, M., Gimpel, K., & Livescu, K. (2015). Deep
multilingual correlation for improved word embeddings. In NAACL.
Faruqui, M., & Dyer, C. (2013). An Information Theoretic Approach to Bilingual
Word Clustering. In ACL.

Multilingual training of embeddings for the sake of better source-language
embeddings:
Suster, S., Titov, I., and van Noord, G. (2016). Bilingual learning of
multi-sense embeddings with discrete autoencoders. In NAACL-HLT.
Guo, J., Che, W., Wang, H., and Liu, T. (2014). Learning sense-specific word
embeddings by exploiting bilingual resources. In COLING.

More broadly, translational context has been explored e.g. in
Diab, M., & Resnik, P. (2002). An unsupervised method for word sense tagging
using parallel corpora. In ACL.